# Shapley Residuals: Quantifying the limits of the Shapley value for explanations

**I. Elizabeth Kumar**
Department of Computer Science
Brown University
Providence, RI 02912
iekumar@brown.edu

**Carlos Scheidegger**
Department of Computer Science
University of Arizona
Tucson, AZ 85721
cscheid@cs.arizona.edu

**Suresh Venkatasubramanian**
Department of Computer Science
Brown University
Providence, RI 02912
suresh@brown.edu

**Sorelle A. Friedler**
Computer Science Dept.
Haverford College
Haverford, PA 19041
sorelle@cs.haverford.edu

## Abstract

Popular feature importance techniques compute additive approximations to nonlinear models by first defining a cooperative game describing the value of different subsets of the model's features, then calculating the resulting game's Shapley values to attribute credit additively between the features. However, the specific modeling settings in which the Shapley values are a poor approximation for the true game have not been well-described. In this paper we utilize an interpretation of Shapley values as the result of an orthogonal projection between vector spaces to calculate a *residual* representing the kernel component of that projection. We provide an algorithm for computing these residuals, characterize different modeling settings based on the value of the residuals, and demonstrate that they capture information about model predictions that Shapley values cannot. Shapley residuals can thus act as a warning to practitioners against overestimating the degree to which Shapley-value-based explanations give them insight into a model.

## 1 Introduction

There have been many recent efforts to quantify the importance of features to a model [19, 4, 1, 15, 12, 13]. Many of these determine the importance through estimating the Shapley value of a game designed to assign importance to sets of features [4, 12, 5, 13, 14, 25]. These Shapley-value-based feature importance methods are used widely in practice [2].

At the same time, there have been increasing concerns that these game theoretic values may not completely capture human or technical notions of feature importance [10, 21, 24]. A particularly salient issue is that users have misconceptions about what Shapley values represent and what actionable information can be gleaned from them [8]. Non-linear complex models, and models built on correlated features, do not have Shapley values that can be interpreted as the effect of a direct intervention [10], e.g., so that increasing a variable value changes the model outcome in a predictable way. The goal of this work is to *quantify* the extent of these concerns and provide a theoretical foundation for understanding the limits of Shapley values.

In this work, we introduce **Shapley Residuals**, vector-valued objects that capture a specific type of quantitative information lost by Shapley values. Shapley residuals can be associated with individual

35th Conference on Neural Information Processing Systems (NeurIPS 2021).

variables, as well as with sets of variables. When the residual of a feature exhibits a large norm, the associated Shapley value should be taken with skepticism: the resulting importance is not just due to the variable acting by itself. On the other hand, if a residual is small, most of the effect of the variable on the model is explainable by the variable acting independently (we make these statements precise in Section 3). The Shapley residual, then, communicates important details about what the explanation actually represents.

To build an intuition for why this is an important problem, consider an algorithm which makes admissions decisions purely on the basis of gender and department: $f(g, d) = g + d - 2dg$, where $g = -1$ if the applicant is male and $g = 1$ otherwise, and there are two departments, represented by $d = 1$ and $d = -1$. In this contrived scenario, the applicant is admitted if $f(g, d) > 0$ (which only happens when $g$ and $d$ have different signs) and is rejected otherwise. Clearly, the admissions decision is affected by gender–yet if each of the two variables are distributed with mean 0, the KernelSHAP values [12] which are supposed to explain the decision $f(1, 1) = 0$ are both 0, since to compute the Shapley value, each features' univariate and interaction influences are averaged together and cancel each other out. In this way, the computation of the Shapley value has implicitly obscured a discriminatory effect, and the corresponding nonzero Shapley residuals would demonstrate that the Shapley values are not telling the whole story.

To more precisely describe what Shapley residuals capture, consider the following two motivating scenarios. First, suppose a practitioner uses Shapley values to determine the effect of *data interventions* on model outcomes. Consider two models $f_1$ and $f_2$. In a real-world scenario, the practitioner will often only have black-box access to such models, and the models will often be significantly more complex. Here, we use these simple models:

$$
\begin{aligned}
f_1(x_1, x_2, x_3) &= x_1 + x_2 + x_3 \\
f_2(x_1, x_2, x_3) &= x_1 + 2x_2 x_3
\end{aligned}
$$

Suppose the practitioner seeks to explain the output $f_1(1, 1, 1) = 3$ or $f_2(1, 1, 1) = 3$, using KernelSHAP to compute local feature importances. For both models, the Shapley values of $x_1$, $x_2$, and $x_3$ are all 1. Despite that, intervening by increasing the value of $x_2$ changes $f_2$ more than increasing the value of $x_1$; in $f_1$, this clearly does not happen. The Shapley residuals for all variables in $f_1$ are zero, indicating that variables in $f_1$ do not interact (as we prove in Section 3). The Shapley residuals for $x_2$ and $x_3$ in $f_2$, on the other hand, are nonzero, while the Shapley residual of $x_1$ is still zero. Finally, the Shapley residual for the set of variables $\{x_2, x_3\}$ is also zero. As we show in Section 3, these statements imply the following behavior for variables of $f_2$: $x_1$ has no interactions with other variables (its residual is zero); $x_2$ and $x_3$ interact with other variables (their residuals are non zero); $x_2$ and $x_3$ only interact *with each other* (the residual of the set $\{x_2, x_3\}$ is zero). Thus, access to Shapley residuals gives warning that intervening on $x_2$ or $x_3$ in $f_2$ could act differently than $x_1$ due to an interaction between $x_2$ and $x_3$.

Table 1: KernelSHAP game for Example 1 - the input $(1, 1, 1)$ to $f(x_1, x_2, x_3) = x_1 + 2x_2 x_3$ where $x_i$ are iid $\mathcal{N}(0, 1)$ features.

| S | Hypercube Coordinate | $v(S)$ Definition for explaining $(1, 1, 1)$ with KernelSHAP | $v(S)$ Value given i.i.d. $x_i \sim \mathcal{N}(0, 1)$ |
|---|---|---|---|
| $\emptyset$ | (0,0,0) | $\mathbb{E}[f(\boldsymbol{x})]$ | 0 |
| $\{x_1\}$ | (1,0,0) | $\mathbb{E}[f(\boldsymbol{x})\|x_1 = 1]$ | 1 |
| $\{x_2\}$ | (0,1,0) | $\mathbb{E}[f(\boldsymbol{x})\|x_2 = 1]$ | 0 |
| $\{x_3\}$ | (0,0,1) | $\mathbb{E}[f(\boldsymbol{x})\|x_3 = 1]$ | 0 |
| $\{x_1, x_2\}$ | (1,1,0) | $\mathbb{E}[f(\boldsymbol{x})\|x_1 = 1, x_2 = 1]$ | 1 |
| $\{x_1, x_3\}$ | (1,0,1) | $\mathbb{E}[f(\boldsymbol{x})\|x_1 = 1, x_3 = 1]$ | 1 |
| $\{x_2, x_3\}$ | (0,1,1) | $\mathbb{E}[f(\boldsymbol{x})\|x_2 = 1, x_3 = 1]$ | 2 |
| $\{x_1, x_2, x_3\}$ | (1,1,1) | $\mathbb{E}[f(\boldsymbol{x})\|x_1 = 1, x_2 = 1, x_3 = 1]$ | 3 |

In the second scenario, consider a data generating distribution where $\alpha$ controls the correlation between two features in $X$ and a regression target $y$:

$$
(X, y) \sim \left( \mathcal{N} \left( (0, 0), \begin{bmatrix} 1 & \alpha \\ \alpha & 1 \end{bmatrix} \right), \langle X, (3, 1) \rangle \right).
$$

We examine a regression model $f(x_1, x_2) = \beta_1 x_1 + \beta_2 x_2$ determined via linear least squares. Assume access to infinitely many IID samples from $(X, y)$, $\beta = (3, 1)$. Suppose a practitioner wanted to explain the output of $f(1, 1) = \beta_1 + \beta_2$, this time using Conditional Expectation SHAP [24]. The Shapley values are $\beta_1 + \alpha(\beta_2 - \beta_1)/2$ for $x_1$ and $\beta_2 + \alpha(\beta_1 - \beta_2)/2$ for $x_2$. When $\alpha \approx 0$, Shapley values correspond to model weights $\beta_1, \beta_2$, and support a (valid) interventional interpretation that changing $x_1$ yields a larger change to the output of $f$ than does $x_2$. However, if $\alpha \approx 1$, Shapley values do not support this interpretation. A practitioner employing Shapley values alone lacks the information to distinguish these scenarios. Shapley residuals provide useful diagnostic information; the norm of the residuals for $x_1$ and $x_2$ is exactly linearly proportional to $\alpha$.

In these simple scenarios, it is clear that Shapley residuals capture, respectively, *variable interactions* and *mismatches between dependent features in the data and independent variables in the model*. As we show in Section 6, these observations apply to real-world scenarios as well.

In summary, we:

- introduce *Shapley residuals* (Section 3), which characterize the limits of Shapley values as explanatory mechanisms for cooperative games,

- study the properties of Shapley residuals both in general and in context of existing formulations for explanatory games (Sections 3, 4 and 5),

- show via a number of experiments that Shapley residuals capture meaningful information for model explanations in realistic scenarios (Section 6),

- discuss the limitations of Shapley residuals themselves (Section 7).

## 2 Background

In this section, we begin by setting up the mathematical definitions and background we'll need for the rest of the paper. To help illustrate these ideas, we'll use the running example from the introduction of function $f = x_1 + 2x_2x_3$; we refer to this as Example 1. We begin by describing Shapley values and cooperative games.

**Games.** A *cooperative* game consists of $d$ players and a *value function* $v : 2^{[d]} \to \mathbb{R}$ where $[d] \triangleq \{1, \ldots, d\}$. The quantity $v(S)$ represents the value of the game for a coalition of players $S \in N \triangleq 2^{[d]}$. Without loss of generality we will assume that $v(\emptyset) = 0$, and that we can identify the game with $v$. Let the space of games be denoted by $\mathcal{G}$.

The Shapley value is a way to fairly allocate the value of the grand coalition $v([d])$ among the players.

**Definition 1** (Shapley values[20]). *The Shapley values $\phi_i(v), i \in [d]$ are the unique values satisfying the properties*

**Efficiency:** $\sum_{i=1}^{d} \phi_i(v) = v([d])$.

**Dummy:** *If $v(S \cup \{i\}) = v(S)$ for all $S \subset [d] \setminus \{i\}$, then $\phi_i(v) = 0$.*

**Symmetry:** *If $v(S \cup \{i\}) = v(S \cup \{j\})$ for all $S \subset [d] \setminus \{i, j\}$, then $\phi_i(v) = \phi_j(v)$.*

**Linearity:** *If $v, v'$ are two games on $d$ players, then $\phi_i(\alpha v + \alpha' v') = \alpha \phi_i(v) + \alpha' \phi_i(v')$.*

Given a model $f(x_1, x_2, ..., x_d)$, the features from 1 to $d$ can be considered players in a game in which the payoff $v$ is some measure of the importance or influence of a subset of features. The Shapley value $\phi_i(v)$ can then be viewed as a fairly attributed "influence" of $i$ on the outcome $v([d])$. In KernelSHAP, for instance, a function's prediction on a certain input given a data distribution is modeled as a game as shown in Table 1.

It will be useful for us to visualize a game as a function over the vertices of a $d$-dimensional hypercube. Each coordinate corresponds to the presence or absence of a certain player, and each vertex corresponds to a subset of players. Specifically, we can think of the set $N$ as the $d$-dimensional hypercube $G = (V = N, E)$ with each vertex labeled by a set $S \subseteq [d]$ and edges between sets $S$ and $S \cup \{i\}$ for all $i \in [d], S$. We depict this interpretation in Figure 1(a).

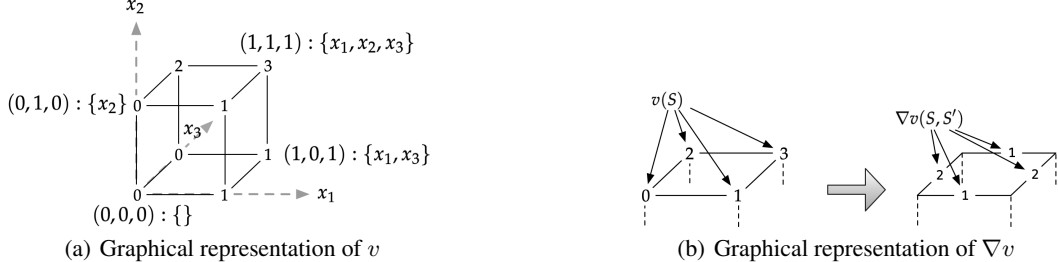

(a) Graphical representation of $v$

(b) Graphical representation of $\nabla v$

Figure 1: Visualizing the game and gradient of the game corresponding to Example 1.

**Gradients on the hypercube.** Let $\mathbb{R}^V$ be the space of functions from $V$ to $\mathbb{R}$ and let $\mathbb{R}^E$ be the space of functions from $E$ to $\mathbb{R}$. In particular, the game $v$ is an element of $\mathbb{R}^V$.

The differential operator $\nabla : \mathbb{R}^V \to \mathbb{R}^E$ is then defined as $\nabla v(S, S \cup \{i\}) = v(S \cup i) - v(S)$ for any $v \in \mathbb{R}^V$. Essentially $\nabla$ is a discrete gradient operator on $G$, mapping functions on vertices to functions on edges (see Figure 1(b)).

We will also define a partial gradient $\nabla_i : \mathbb{R}^V \to \mathbb{R}^E$:

$$\nabla_i u(S, S \cup \{j\}) = \begin{cases} u(S \cup j) - u(S) & i = j \\ 0 & \text{otherwise} \end{cases}$$

Intuitively, $\nabla_i$ evaluates a gradient for edges corresponding to the insertion of $i$, and takes the value 0 everywhere else. On the hypercube, only edges on the $i$th axis of $\nabla_i v$ will take a nonzero value. See the Edge Space portion of Figure 2(a) for an illustration of this procedure on the running example.

## 2.1 Geometric characterization of Shapley values

A geometric interpretation of Shapley values dates back at least to Kleinberg and Weiss [9], showing they can be expressed in terms of projections from the space of games to the space of cooperative games with independently contributing players. A key advance was made by Stern and Tettenhorst [22], building on earlier work by Candogan et al. [3] who proposed viewing the game as a scalar function defined on the hypercube and studying its discrete gradient. To understand this advance, we first introduce a special class of games.

**Inessential games** Let $\mathcal{I}$ denote the space of games $v$ such that for all $S \subseteq [d]$, $v(S) = \sum_{i \in S} v(\{i\})$. $\mathcal{I}$ is called the space of *inessential games*. Intuitively an inessential game is one in which the player interactions are simple and additive: every player adds a fixed value $v(\{i\})$ to a coalition $S$ independent of the composition of $S$. Inessentiality is a key feature of what makes Shapley values attractive for feature importance – if each contribution is fixed and combines additively, we have a natural interpretation for how much each feature contributes to the overall model output. Specifically, if a game is inessential, it then follows that the Shapley value for player $i$ is $v(\{i\})$. In our running example using KernelSHAP, this is $\mathbb{E}[f(x)|x_i = 1]$, the contribution (averaged over other variables) of the variable $x_i$.

In general though, a game might not be inessential. The key insight of Stern and Tettenhorst [22] was to express inessentiality of games in terms of gradients on the hypercube.

**Proposition 1** ([22, Prop 3.3]). *The game $v$ is inessential if and only if for each $i \in [d]$ there exists $v_i \in \mathbb{R}^V$ such that $\nabla_i v = \nabla v_i$.*

The main result by Stern and Tettenhorst [22] is a decomposition of an arbitrary game $v$ into games that are "close to being inessential" and allow extraction of Shapley values. If $v$ is not inessential, we cannot be sure to find $v_i$ such that $\nabla_i v = \nabla v_i$, but we can find the "closest" such $v_i$ as the solution to the least squares problem

$$\min_{x \in \mathbb{R}^V, x(\emptyset)=0} \|\nabla x - \nabla_i v\|$$

**Theorem 1** (Stern and Tettenhorst [22]). *Given a game $v$, let $v_i$ be defined as above. Then*

    *1.* $\sum v_i = v$

2. *If $v(S \cup \{i\}) = v(S)$ for all $S \subset [d]$, then $v_i = 0$*

3. *For any $\alpha, \alpha' \in \mathbb{R}$ and games $v, v'$, $(\alpha v + \alpha' v')_i = \alpha v_i + \alpha' v'_i$*

4. *If $\pi$ is a permutation of $[d]$ and $\pi \circ v$ is the game $\pi \circ v(S) = v(\pi(S))$, then $(\pi \circ v)_i = v_{\pi(i)}$*

Consider the mapping $\phi(v)(S) = \sum_{i \in S} v_i([d])$. The above result implies this is a Shapley mapping and therefore $\phi_i(v) = v_i([d])$ are the Shapley values of $v$. We illustrate the construction in Figure 2(a).

## 3 Shapley Residuals

The inessentiality of a game is inextricably linked to the meaningfulness of Shapley values for the reasons given above. The idea we explore now is the converse: can the *degree* to which a game is not inessential provide insights into where Shapley values are not able to capture feature influence?

By the fundamental theorem of linear algebra, we can write

$$\nabla_i v = \nabla v_i + r_i$$

where $r_i$ is orthogonal to $\nabla v_i$. This allows us to interpret $r_i$ (a vector with one value for each edge of the hypercube) as a measure of *deviation from inessentiality*, because by Proposition 1, this vector is identically 0 if and only if the game is inessential.

We can generalize these ideas further to subsets of players. We begin with a generalized notion of inessentiality:

**Definition 2.** *The game $v$ is inessential* relative to $S$ if $v(C) = v(S) + v(C \setminus S)$ *for all $S$ and $C$ such that $S \subset C \subset [d]$.*

That is, each coalition containing $S$ obtains a value equal to the subcoalition $S$ working separately from $C \setminus S$; in this sense, inessentiality with respect to $S$ can speak to the lack of interactions between $S$ and its complement. In addition, inessentiality relative to a single player $i$ is the same as inessentiality relative to the singleton set $\{i\}$.

Next, we generalize the notion of a partial derivative.

**Definition 3.** *For a subset $S \subset [d]$, let $\nabla_S \colon \mathbb{R}^V \to \mathbb{R}^E$ be the operator $\nabla_S = \sum_{i \in S} \nabla_i$, or*

$$\nabla_S u(C, C \cup \{j\}) = \begin{cases} \nabla u\,(C, C \cup \{i\}) & \text{if } i = j \text{ and } i \in S, \\ 0 & \text{otherwise.} \end{cases}$$

We can now prove a result similar to Proposition 1 for relative inessentiality.

**Proposition 2.** *The game $v$ is inessential* relative to $S$ if and only if there exists $v_S \in \mathbb{R}^V$ such that $\nabla_S v = \nabla v_S$.

To understand the limits of Shapley values, we propose to quantify the degree of deviation from inessentiality with the following definition:

**Definition 4** (Shapley Residuals). *We call $r_i = \nabla_i v - \nabla v_i$ the* Shapley Residual *of player $i$. Analogously, $r_S = \sum_{i \in S} r_i$ is the* Shapley Residual *of set $S$.*

Shapley Residuals are a novel diagnostic tool for feature importance, and enjoy a number of relevant properties.

**Proposition 3.** *If $v$ is inessential, then $v$ is inessential relative to all $i \in [d]$ and all subsets $S \subset [d]$. If $v$ is inessential with respect to each player of $i, j, \ldots z$ then $v$ is inessential relative to the set $\{i, j, \ldots, z\}$.*

The proof of this proposition is in the appendix. The following corollaries are straightforward.

**Corollary 1.** *$v$ is inessential iff $r_i = 0$ for each $i \in [d]$.*

**Corollary 2.** *$v$ is inessential relative to $S$ iff $r_S = \sum_{i \in S} r_i = 0$.*

This allows us to interpret $\sum_{i \in N} ||r_i||^2$ as the deviation from inessentiality of $v$ and $||\sum_{i \in S} r_i||^2$ as the deviation from inessentiality of $v$ relative to $S$.

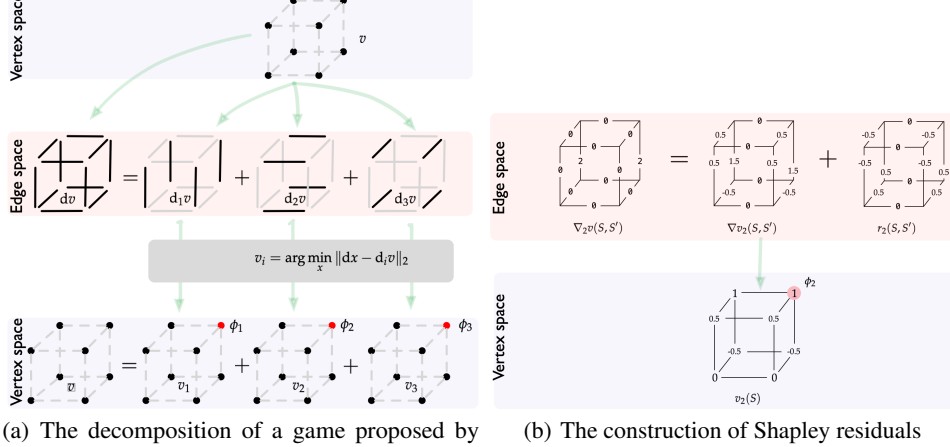

(a) The decomposition of a game proposed by Stern and Tettenhorst [22]

(b) The construction of Shapley residuals

Figure 2: Visualizing the decomposition of a game and its residuals.

In this paper we will focus on the computation and evaluation of residuals with respect to individual players i.e $r_S$ for $S = \{i\}$. Figure 2(b) illustrates the construction of residuals. Algorithm 1 describes how to compute residuals.[1]

---

**Algorithm 1** Exactly calculate the $i$th Shapley value and Shapley residual of $v$

---

Compute $\nabla_i v$
Solve $v_i = \text{argmin}_{x \in \mathbb{R}^V} ||\nabla_i v - \nabla x||_2^2$
Compute $\nabla v_i$
Return Shapley residual $r_i = \nabla_i v - \nabla v_i$
Return Shapley value $\phi_i = v_i(S) - v_i(\emptyset)$ where $S$ is the set of all players

---

## 4 Feature Importance, Inessentiality and Residuals

We have established that the norm of the residual $r_i$ characterizes the degree to which the value function $v$ is not inessential with respect to the player $i$. We now show how to interpret this when attributing feature importance via Shapley values for two popular methods. As has been noted, the different methods for Shapley value-based explanation (whether local or global) all reduce to a specific choice for the game $v$, at which point the Shapley values of $v$ are estimated and returned [24, 10, 16].

The definitions of *Shapley sampling values* [23], as well as *SHAP values* [12], are derived from defining $v$ as the *conditional* expected model output on a data point when only the features in $S$ are known: $v_{f,x}^{Cond}(S) = E[f(\boldsymbol{X})|\boldsymbol{X}_S = \boldsymbol{x}_S]$ We call this Conditional Expectation SHAP after Sundararajan and Najmi [24].

The Interventional SHAP value function, which defines KernelSHAP, is derived from defining $v$ by taking an expectation of $f$ over the joint distribution of $\bar{S}$ while fixing the feature values from $S$: $v_{f,x}^{Int}(S) = E[f([\boldsymbol{x}_S, \boldsymbol{X}_{\bar{S}}])]$ Notably, the two values are the same if the features in $\bar{S}$ are independent from those in $S$.

We will show that the residual $r_S$ captures the degree to which *interactions* between the features in $S$ and its complement arise in the model or in the data, depending on which form of Shapley-based feature importance is used to define the value function $v$.

---

[1]We can take an unconstrained minimum here and subtract $v_i(\emptyset)$ at the end because adding a constant value to $v$ does not change $\nabla v$.

### 4.1 Inessentiality and Interactions

**Interventional SHAP**  Recall the problem of explaining two models where $f_1(x_1, x_2, x_3) = x_1 + x_2 + x_3$ and $f_2(x_1, x_2, x_3) = x_1 + 2x_2x_3$. Note that in the first model all three variables contribute independently to the model output, whereas in the second model the variables $x_2$ and $x_3$ interact in their contribution. We can compute the associated residuals $r_1, r_2, r_3$ and their norms for these two models. For the first one, all residuals are identically zero. However, in the second model if $x_2$ and $x_3$ are nonzero for a certain input, they will have a nonzero residual. In other words, *the residual captures feature interactions in the model*. Our first result, which we prove in the appendix, shows that this intuition can be made precise.

**Lemma 1.** *Let $f : \boldsymbol{X} = \{X_1, X_2, ..., X_d\} \rightarrow Y$ be a multivariate function. Suppose $f$ can be decomposed as $f(\boldsymbol{x}) = g(\boldsymbol{x}_S) + h(\boldsymbol{x}_{\bar{S}})$, for some functions $g : \{X_j : j \in S\} \rightarrow Y$ and $h : \{X_j : j \notin S\} \rightarrow Y$. Let $\boldsymbol{z} = \{z_1, z_2, ..., z_n\} \in \boldsymbol{X}$. Then $v_{f,z}^{Int}$ is relatively inessential with respect to the set $S$.*

This is important because if the model really does decompose additively for a certain variable $i$, the practitioner understands what to expect when variable $i$ is perturbed. The Interventional Shapley residuals thus quantify the extent to which the SHAP values must be augmented with more information to capture interaction effects in the model.

**Conditional Expectation SHAP**  As the residual for Interventional SHAP can be thought of as detecting feature interactions in a model, the residuals of Conditional Expectation SHAP can detect feature interactions in the data. Let $\boldsymbol{X} \sim \mathcal{N}([0,0]^T, \Sigma)$ for $\Sigma = \begin{bmatrix} 1 & \alpha \\ \alpha & 1 \end{bmatrix}$, and let $Y = f(\boldsymbol{X}) = \beta^T \boldsymbol{X}$ (note that ordinary least squares will recover $f$ in the limit of infinite data). Given input $x_1, x_2$, the SHAP values of $f$ are $\phi_1 = \beta_1 x_1 + \alpha \frac{\beta_2 x_1 - \beta_1 x_2}{2}$ $\phi_2 = \beta_2 x_2 + \alpha \frac{\beta_1 x_2 - \beta_2 x_1}{2}$. That is to say, they are linearly dependent on the correlation between the two variables. In particular, consider explaining the input $[1, 1]$ to the function $x_1 + 3x_2$; the SHAP values are $\phi_1 = 1 + \alpha$ and $\phi_2 = 3 - \alpha$ and the residuals are both $2\alpha$. Notably, as interaction between variables increases in the data (measured by $\alpha$), the residual increases and the SHAP values deviate further and further from the coefficients of the actual model. We can make this intuition precise.

**Lemma 2.** *Let $f : \boldsymbol{X} = \{X_1, X_2, ..., X_d\} \rightarrow Y$ be a multivariate function. Suppose $f$ can be decomposed as $f(\boldsymbol{x}) = g(\boldsymbol{x}_S) + h(\boldsymbol{x}_{\bar{S}})$, for some functions $g : \{X_j : j \in S\} \rightarrow Y$ and $h : \{X_j : j \notin S\} \rightarrow Y$. Let $\boldsymbol{z} = \{z_1, z_2, ..., z_n\} \in \boldsymbol{X}$. Suppose further that all $X_j : j \in S$ are distributed independently from all $X_j : j \notin S$. Then $v_{f,z}^{Cond}$ is relatively inessential with respect to set $S$.*

The residual on Conditional Expectation SHAP thus quantifies the extent to which an interpretation of the SHAP values can be interpreted as interventional, because depending on the causal structure of the data, correlated features could imply that perturbing a feature $i$ could result in the perturbation of a different feature as well.

**Inspecting Shapley residuals in practice**  Shapley residuals are vectors in the same space as gradients, and are generally high-dimensional entities; a full study of their properties remains an important topic for future work. The characterization in this section shows that the *norm* of the residual vectors captures important limitations of Shapley values. Thus, our experiments use the *scaled norm* of the residual vectors, defined to be the norm of the residual vector divided by the norm of the discrete gradient vector. Normalized residuals make them easier to compare across experiments.

## 5  Relationship with Other Interaction Indices

[25], similarly recognizing that Shapley values lose information about interactions, proposed Shapley-Taylor Interaction Indices, a generalization of Shapley values which attributes influence among interaction terms. Specifically, the Shapley-Taylor explanation for $x$ of order $k$ assigns values $I_S^k$ to subsets of features $S$ of size $|S| \leq k$ such that $\sum I_S^k = f(x)$. The terms for the subsets for which $|S| < k$ represent a discrete Taylor series around $v(\emptyset)$. When $|S| = k$, $I_S^k$ is defined similarly to the Shapley value: a discrete derivative averaged over permutations.

Our residuals capture fundamentally different information about interactions than Shapley-Taylor. Consider some subset $S$ for which $|S| < k$. Our residual $r_S$ is 0 when the marginal value of adding $S$ to a coalition $W$ is constant with respect to sets for which $W \cap S = \emptyset$; in other words, it is about the presence or absence of interactions of $S$ with other variables. The Shapley-Taylor interaction index $I_S^k$, on the other hand, is 0 when $v(S)$ can be inferred from the values of $v(W)$ for $\{W \subset S\}$. The Taylor terms of the Shapley-Taylor explanation thus capture information about how the players in $S$ interact with each other *when no other variables are involved*. For instance, if for a certain game $v(\{i\}) + v(\{j\}) = v(\{i,j\})$, this means that the term $v(\{i,j\})$ provides no interaction information about the two players, and $I_{\{i,j\}}^k$ for explanation sizes $k > 2$ will be 0.

However, the Taylor indices for a coalition $S$ say nothing about whether the variables within $S$ interact *once a player outside of $S$ is involved.* Consider a three-player game between $a, b$, and $c$, where $v(\{a\}) + v(\{b\}) = v(\{a,b\})$ and $v(\{a\}) + v(\{c\}) = v(\{a,c\})$; this would make $I_{\{a,b\}}^k$ and $I_{\{a,c\}}^k$ equal to 0, implying that $a$ and $b$ do not interact, and $a$ and $c$ do not interact. But it could be that $v$ is *not* relatively inessential with respect to $a$. If $v(\{a,b,c\}) - v(\{b,c\}) \neq v(\{a,c\}) - v(\{c\})$, then $a$'s relative contribution with respect to $c$ changes once $b$ is involved. This constitutes an interaction between $a$ and $c$ that is not described by the Shapley-Taylor index for $\{a,c\}$, but is rather captured by the third-order interaction of $\{a,b,c\}$. In this scenario, our residuals would show $r_a \neq 0$, alerting us to the fact that $a$ interacts with $\{b,c\}$; additionally, it would have $r_{\{a,b\}} \neq 0$, alerting us to the fact that $\{a,b\}$ interacts with $c$.

In general, since $r_i$ captures information about *all* of $i$'s interactions, we can state the following connection between the two Shapley extensions:

**Lemma 3.** *Given subset $S, |S| < k$, if $\exists i \in S$ s.t $r_i = 0$, then the Shapley-Taylor index $I_S^k = 0$.*

We have focused our attention on Shapley-Taylor interaction indices because of their proposed use for explanations. It should be noted that they (as well as the Shapley interaction index proposed in [13]) are special cases of a general class of interaction indexes investigated in a long line of work starting with [17] and surveyed in [6] (including the Grabisch-Roubens[7] Shapley and Banzhaf interaction indices). All of these differ from Shapley residuals – the latter are meant to represent the information lost when computing *singleton* Shapley values, not their higher-dimensional extensions, which are based on a different notion of a derivative.

## 6   Experiments

Having theoretically justified Shapley residuals in previous sections, we now focus on illustrating what these residuals can help us understand about models on a real-world dataset. Throughout, we use our own implementation of KernelSHAP to calculate the exact Shapley values and residuals (see Algorithm 1 in Section 3).[2] Some additional experiments can be found in the appendix.

**On comparisons to other feature importance methods**   We note that Shapley residuals are not a feature importance evaluation method, nor are they an "explanation method" in and of themselves. Rather, they are a quantification of the (valuable) information *lost* by Shapley values. A direct comparison of different feature influence evaluation methods makes sense when there is a clear objective to compare against. Such an objective doesn't really exist here. Rather, we choose to provide an *internal* validation that lays out the mathematical foundation on which the method rests. This allows a user to decide the context in which to employ one method or another. For example, as we discussed in Section 5, Shapley-Taylor indices and Shapley residuals appear to capture different kinds of interactions that are potentially of interest to a user. There is no meaningful way to compare them in a vacuum because one is not "better" than another.

**Variable Interactions in Occupancy Detection**   Consider the Shapley values and residuals for an occupancy detection dataset[3] (20,560 instances) used to predict whether an office room is occupied. The 7 attributes include a date stamp for an hour and day of the week. A decision tree model with maximum depth 3 is trained on 75% of the data using the features `light` and `hour`. When

---

[2]Code is provided in the supplementary material
[3]https://archive.ics.uci.edu/ml/datasets/Occupancy+Detection+

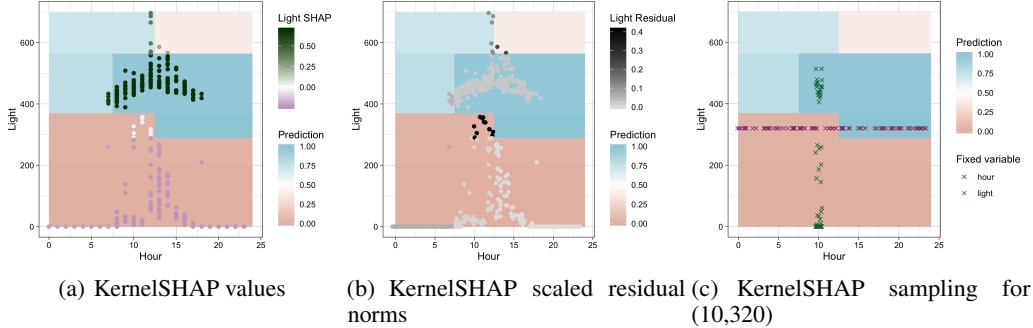

(a) KernelSHAP values     (b) KernelSHAP scaled residual (c) KernelSHAP sampling for norms     (10,320)

Figure 3: Shapley values and residuals on a decision tree for the Occupancy Detection task

evaluated on the remaining test set, the ROC-AUC for this decision tree is $0.991$. We then calculate the Shapley values and residuals (using 50 randomly sampled background rows from the test set) for 1000 randomly sampled test instances. The results for the variable "light" are shown in Figure 3.

The reason that the cluster of points in the middle has a high residual is illustrated in Figure 3(c). Calculating the expected prediction while fixing a light value of 320, unlike most other possible values, results in a mix of low and high predictions. These average to $0.4$, while both the overall expectation and particular prediction for occupancy probability for those points are $0.25$.

$$\mathbb{E}[f(H, L)] = .24 \xrightarrow{+.01} \mathbb{E}[f(10, L)] = .25$$
$$\Big\downarrow{\scriptstyle +.16} \qquad\qquad\qquad\qquad \Big\downarrow{\scriptstyle -.24}$$
$$\mathbb{E}[f(H, 320)] = .40 \xrightarrow{-.39} f(10, 320) = .01$$

Figure 4: Geometric representation of the KernelSHAP game for $f(10, 320)$, where arrows to the right indicate inclusion of the `light` feature and arrows down indicate inclusion of the `hour` feature.

Specifically, the KernelSHAP game for $f(H, L) = P(\text{occupant} = T)$ for $L = 320$ and $H = 10$ is shown in Figure 4. $L = 320$ is a positive indicator of occupancy if H is unknown (+.16) but is a "negative" indicator of occupancy is H is known to be 10 (-.24), due to the interactions in the model in this area of the feature space. The light Shapley value is close to 0 for points in this range, then, because it is the average of a positive and negative number – not because it is of "low importance" – and the non-inessentiality of this feature is what is being captured by the residual.

## 7 Limitations and Future Work

**Usability considerations**    Our motivation for this work is to contribute further to the theoretical foundation of Shapley-value-based feature importance measures and, critically, to introduce Shapley residuals to quantify missing importance. Our goal is that residuals be a *warning* attached to specific Shapley values and thus alert practitioners to model complexities and importances that have previously gone unattended. Further research is needed to investigate whether these residuals can be effectively utilized by humans to make better decisions about their models. An empirical, human-centered investigation is critical because, like Shapley values themselves, the meaning of these residuals may be hard for practitioners to understand, and therefore errors in the interpretation of these residuals may cause unanticipated negative consequences.

**Performance considerations**    SHAP implementations provide a partial evaluation of the game vector [12, 11], which provides analysts with results even in high-dimensional settings. Unfortunately, there is no assumption-free provable bound on the relationship between partially evaluated game vectors and actual Shapley values. Our goal here is more precisely characterize the information not conveyed by Shapley values and so we always compute the full vector. Thus, the runtime is ultimately exponential in the number of variables to analyze. This currently limits the number of

variables for which Shapley residuals can be practically computed to 20 to 30 (with corresponding vectors of length between a million and a billion elements). Since the derivative operator is sparse and well-conditioned, the least squares problem is efficiently solved by the LSQR method [18]. Still, in future work, we hope to efficiently identify whether a particular residual is nonzero, and approximate properties of residuals which capture the entirety of non-linear interactions of a particular feature.

**Conclusion** A goal in interpretable machine learning, and within Shapley-value-based feature importance, is to give a rigorous theoretical foundation to interpretability notions so that practitioners can better understand the impacts of their models. This is especially important in contexts where models make high-stakes decisions about people, e.g., via criminal risk assessments and interview screening algorithms. We believe people have the right to understand those decisions, and particularly which features were important for the decision. Putting such feature importance measurements on solid theoretical grounds is important for the validity of these feature importance claims. Their validity is an important part of the ethics of algorithms as societal interventions.

# 8 Acknowledgements

This research was funded in part by the NSF under grants DMR 1928882, IIS 1955162, and IIS 1956286, and by the DARPA SD2 program.

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
