# A Proofs

## A.1 Section 3

Proposition 3:

*Proof.* Stern and Tettenhorst define $C_{\sigma,i} = \{j \in [d] : \sigma(j) < \sigma(i)\}$. We will need the slight extension to sets $C_{\sigma,S} = \bigcap_{i \in S} C_{\sigma,i}$. For any permutation $\sigma$ of $[d]$, which defines a path from $\emptyset$ to $[d]$, the marginal value contributed by subset $S$ along this path is

$$\sum_{j \in [d]} \nabla_S v(C_{\sigma,j}, C_{\sigma,j} \cup \{j\}) = \sum_{i \in S} \nabla_i v(C_{\sigma,j}, C_{\sigma,j} \cup \{j\})$$

$$= \sum_{i \in S} v(C_{\sigma,i} \cup \{i\}) - v(C_{\sigma,i}),$$

which can also be interpreted as a discrete "line integral" of $\nabla_S v$ along the path.

We need to show that the different paths that can be taken through the nonzero entries of $\nabla_S$ sum to the same value. But if $\nabla_S v \in \mathcal{R}(\nabla)$, then the right-hand side of the result in the sum defined in 3.1$_\mathbf{S}$ telescopes to $v(C_{\sigma,S} \cup S) - v(C_{\sigma,S})$, since $\nabla_S v \in \mathcal{R}(\nabla)$ implies path independence inside $S$. As a result, the marginal value $v(S \cup C) - v(S)$ is the same for all coalitions $C \subset N \setminus S$. Taking $C = \emptyset$, we see that this value is precisely $v(S)$, and we conclude that $v$ is inessential relative to $S$. Conversely, suppose that $v$ is inessential relative to $S$, and define the game

$$v_S(C) = \begin{cases} v(S \cap C) & \text{if } S \cap C \neq \emptyset, \\ 0 & \text{if } S \cap C = \emptyset. \end{cases}$$

It follows immediately that $\left(\sum_{i \in S} \nabla_i\right) v = \nabla v_S \in \mathcal{R}(\nabla)$, which completes the proof. $\square$

## A.2 Section 4

Lemma 1:

*Proof.* Using the linearity of expectation, we can rewrite this game as

$$v_{f,z}^{Int}(T) = E[f([\mathbf{z}_T, \mathbf{X}_{\bar{T}}])] = \begin{cases} g(\mathbf{z}_S) + \mathbb{E}[h([\mathbf{z}_{T \setminus S}, \mathbf{X}_{\bar{T}}])] & S \subseteq T \\ \mathbb{E}[g(\mathbf{X}_S)] + \mathbb{E}[h([\mathbf{z}_T, \mathbf{X}_{\bar{T} \setminus S}])] & T \cap S = \emptyset \end{cases}$$

Now we can write the nonzero elements of the partial derivative $\nabla_S v_{f,z}^{Kernel}$ as

$$
\begin{aligned}
v_{f,z}^{Int}(T \cup S) - v_{f,z}^{Int}(T) =& g(\mathbf{z}_S) + \mathbb{E}[h([\mathbf{z}_{(T \cup S) \setminus S}, \mathbf{X}_{T \bar{\cup} S}])] \\
& - \left(\mathbb{E}[g(\mathbf{X}_S)] + \mathbb{E}[h([\mathbf{z}_T, \mathbf{X}_{\bar{T} \setminus S}])]\right) \\
=& g(\mathbf{z}_S) + \mathbb{E}[h([\mathbf{z}_T, \mathbf{X}_{\bar{T} \setminus S}])] \\
& - \left(\mathbb{E}[g(\mathbf{X}_S)] + \mathbb{E}[h([\mathbf{z}_T, \mathbf{X}_{\bar{T} \setminus S}])]\right) \\
=& g(\mathbf{z}_S) - \mathbb{E}[g(\mathbf{X}_S)]
\end{aligned}
$$

regardless of $T$, as long as $T \cap S = 0$. $\square$

Lemma 2:

*Proof.* Using the linearity of expectation, we can rewrite this game as

$$
\begin{aligned}
v_{f,z}^{Cond}(T) &= E[f(\boldsymbol{X})|\boldsymbol{X}_T = \boldsymbol{z}_T] \\
&= \mathbb{E}[g(\boldsymbol{X}_S)|\boldsymbol{X}_T = \boldsymbol{z}_T] + \mathbb{E}[h(\boldsymbol{X}_{\bar{S}})|\boldsymbol{X}_T = \boldsymbol{z}_T] \\
&= \begin{cases} g(\boldsymbol{z}_S) + \mathbb{E}[h(\boldsymbol{X}_{\bar{S}})|\boldsymbol{X}_{T \setminus S} = \boldsymbol{z}_{T \setminus S}] & S \subseteq T \\ \mathbb{E}[g(\boldsymbol{X}_S)] + \mathbb{E}[h(\boldsymbol{X}_{\bar{S}})|\boldsymbol{X}_T = \boldsymbol{z}_T] & T \cap S = \emptyset \end{cases}
\end{aligned}
$$

Now we can write the nonzero elements of the partial derivative $\nabla_S v_{f,z}^{Cond}$ as

$$
\begin{aligned}
v_{f,z}^{Cond}(T \cup S) - v_{f,z}^{Cond}(T) =& g(\boldsymbol{z}_S) + \mathbb{E}[h(\boldsymbol{X}_{\bar{S}})|\boldsymbol{X}_T = \boldsymbol{z}_T] \\
& - (\mathbb{E}[g(\boldsymbol{X}_S)] + \mathbb{E}[h(\boldsymbol{X}_{\bar{S}})|\boldsymbol{X}_T = \boldsymbol{z}_T]) \\
=& g(\boldsymbol{z}_S) - \mathbb{E}[g(\boldsymbol{X}_S)]
\end{aligned}
$$

regardless of $T$, as long as $T \cap S = 0$.

$\square$

## A.3 Section 5

Lemma 3

*Proof.* Let $i \in S$. Suppose $k > |S|$. Then

$$
\begin{aligned}
I_S^k &= \delta_S(\emptyset) \\
&= \sum_{W \subseteq S} (-1)^{|W|-|S|} v(W) \\
&= \sum_{k=1}^{|S|} \sum_{\substack{W \subseteq S \\ |W|=k}} (-1)^{k-|S|} v(W) \\
&= \sum_{k=1}^{|S|} (-1)^{k-|S|} \sum_{\substack{W \subseteq S \\ |W|=k}} v(W)
\end{aligned}
$$

Notice that

$$
\begin{aligned}
\sum_{\substack{W \subseteq S \\ |W|=k}} v(W) &= \left( \sum_{\substack{W \subseteq S \\ i \notin W \\ |W|=k}} v(W) + \sum_{\substack{W \subseteq S \\ i \in W \\ |W|=k}} v(W) \right) \\
&= \left( \sum_{\substack{W \subseteq S \\ i \notin W \\ |W|=k}} v(W) + \sum_{\substack{T \subseteq S \\ i \notin W \\ |W|=k-1}} v(T \cup i) \right)
\end{aligned}
$$

Further,

$$\sum_{k=1}^{|S|}(-1)^{k-|S|}\sum_{\substack{T\subseteq S\\i\notin W\\|W|=k-1}}v(T\cup i)$$

$$=\sum_{j=0}^{|S|-1}(-1)^{j-|S|+1}\sum_{\substack{T\subseteq S\\i\notin W\\|W|=j}}v(T\cup i)$$

And

$$\sum_{k=1}^{|S|}(-1)^{k-|S|}\sum_{\substack{W\subseteq S\\i\notin W\\|W|=k}}v(W)$$

$$=\sum_{k=0}^{|S|-1}(-1)^{k-|S|}\sum_{\substack{W\subseteq S\\i\notin W\\|W|=k}}v(W)$$

$$=\sum_{k=0}^{|S|-1}(-1)^{k-|S|+1}\sum_{\substack{W\subseteq S\\i\notin W\\|W|=k}}-v(W)$$

So,

$$\sum_{k=1}^{|S|}(-1)^{k-|S|}\sum_{\substack{W\subseteq S\\|W|=k}}v(W)$$

$$=\sum_{k=0}^{|S|-1}(-1)^{k-|S|+1}\sum_{\substack{W\subseteq S\\i\notin W\\|W|=k}}v(W\cup i)-v(W)$$

If $r_i = 0$, this is then equal to

$$\sum_{k=0}^{|S|-1}(-1)^{k-|S|+1}\sum_{\substack{W\subseteq S\\i\notin W\\|W|=k}}v(i)$$

Note that the number of subsets $W \subseteq S$ such that $i \notin S$ and $|W| = k$ is exactly $\binom{|S|-1}{k}$. Thus we can rewrite our expression to

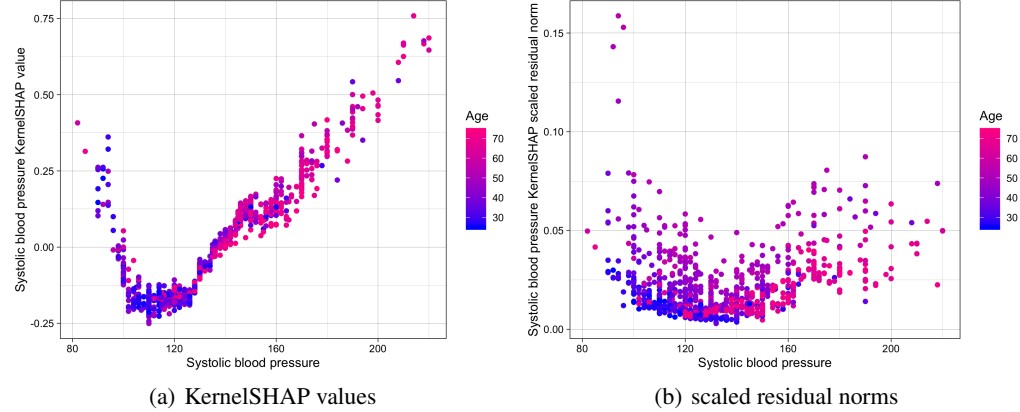

| (a) KernelSHAP values | (b) scaled residual norms |

Figure 5: Shapley values and residuals on an XGBoost mortality model for Systolic blood pressure and age.

$$\sum_{k=0}^{|S|-1} (-1)^{k-|S|+1} \sum_{\substack{W \subseteq S \\ i \notin W \\ |W|=k}} v(i)$$

$$= \sum_{k=0}^{|S|-1} (-1)^{k-|S|+1} \binom{|S|-1}{k} v(i)$$

$$= (-1)^{1-|S|} v(i) \sum_{k=0}^{|S|-1} (-1)^k \binom{|S|-1}{k}$$

Finally we can observe that $\sum_{k=0}^{|S|-1} (-1)^k \binom{|S|-1}{k} = 0$, which means $I_S^k = 0$.

□

## B  Experiments

### B.1  NHANES

The NHANES data, made available via the SHAP package[4], contains 9,932 instances of right-censored mortality data. We use the preprocessing of the data from the SHAP package and train an XGBoost Cox survival model with 5000 estimators on 7 variables ('Age', 'Diastolic BP', 'Sex', 'Systolic BP', 'Poverty index', 'White blood cells', and 'BMI'). The resulting Harrell's C-statistic on the test set is 0.825. We then explain its marginal predictions on 1000 randomly chosen test instances with KernelSHAP on 100 background samples. The resulting KernelSHAP values and residuals for some features are given in Figures 5 and 6.

Considering the feature importance of blood pressure (Figure 5(a)), we find (as also found in Lundberg and Lee [12]) that as blood pressure increases, the importance of blood pressure to mortality also increases, and this effect is correlated with an increase in age. Examining the residuals (Figure 5(b)) additionally allows us to see that there is a missing importance associated with blood pressure for middle aged people across all blood pressure readings, perhaps indicating that blood pressure acts in combination with other variables to impact mortality for this age range.

KernelSHAP feature importance of sex on mortality within this model (Figure 6(a)) shows that being male is, consistently across ages, more predictive of dying than being female. However, the

---

[4]`https://slundberg.github.io/shap/notebooks/NHANES%20I%20Survival%20Model.html`

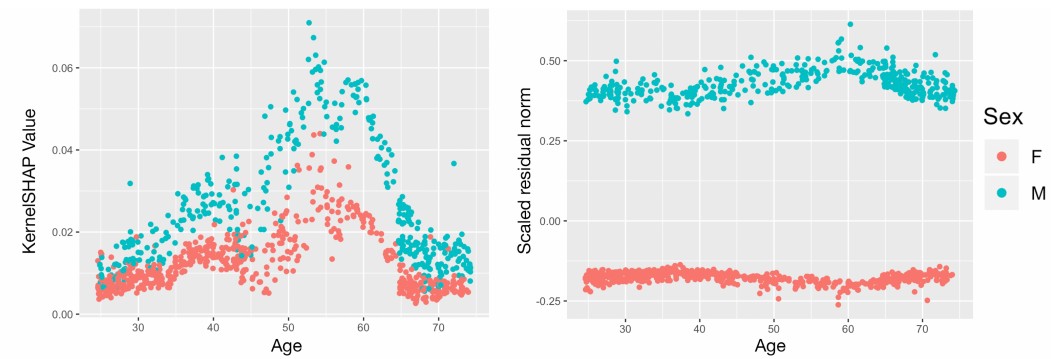

Figure 6: Shapley values and residuals on an XGBoost mortality model for age and sex.

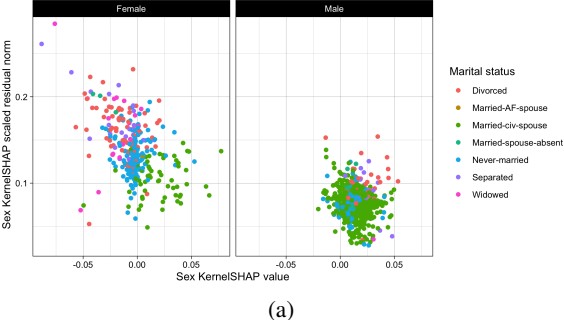

(a)

Figure 7: Shapley values and residuals on a random forest income model.

corresponding residuals (Figure 6(b)) show that middle aged men may have many other interacting and contributing factors for predicting mortality. These two residual charts taken together (Figures 5(b) and 6(b)) may indicate that blood pressure, sex, and age interact within the model to increase the importance of both sex and blood pressure for mortality predictions of middle aged men.

## B.2   Adult Income

The Adult Income dataset[5] contains 48,842 instances of people's census information from 1994, including 14 attributes describing their education, job, marital status, etc., and with the goal of predicting whether the person makes more or less than $50,000 per year. We preprocess the data by removing rows with missing values and train a random forest with 10 trees on all the variables (except `fnlwgt`) on 80% of the data. The ROC-AUC of the model evaluated on the remaining 20% is .857. We calculate the Shapley values and residuals using 1000 test instances and KernelSHAP with 25 background samples. The results for features sex and marital status are shown in Figure 7.

For both men and women, the distribution of Shapley values indicating the importance of sex to the income prediction model is close to a Shapley value of 0. However, in addition to the Shapley values for women having a larger variance, we see with Shapley residuals that residuals for some women are also much higher than those for men. Specifically, while essentially all men have low residuals, essentially only women who are also married to civilian non-absent spouses have low residuals. This indicates that sex and marital status interact in more complex ways with the income prediction model for women than they do for men.

---

[5] http://archive.ics.uci.edu/ml/datasets/Adult