# OpenReview forum: "Shapley Residuals: Quantifying the limits of the Shapley value for explanations"
_NeurIPS.cc/2021/Conference — NeurIPS 2021 Poster_

### Official Review · Reviewer_6Qs2 · 2021-07-15

**Rating:** 7
**Confidence:** 3

**Summary:**

In a cooperative game in characteristic function form, the value function $v$ maps from all subsets of agents, $N = {1, \ldots, n}$, producing an image with $2^N$ elements.  Shapley's value (perhaps the most widely used measure of feature importance in handling ML's attribution problem) ascribes a number to each agent, for a set of $N$ values.

As $N < 2^N$, Shapley's value does not - in general - fully encode the game.  An exception occurs when the game is _inessential_, so that $v(S) = \sum_{i \in S} v(i)$.  Specifically, when features _interact_, Shapley's value may be restrictive.

This paper builds on Stern and Tettenhorst's 2019 _Games and Economic Behavior_ paper, which provided a way of assessing a game's deviation from _inessentiality_, by means of Shapley residuals, $r_i$.

Definition 4 defines _Shapley residuals_, $r_S$, for sets of agents/features.  A series of results (Proposition 1, Corollaries 1, 2) then allow the norms of $r_S$ to be interpreted as a deviation from essentiality and - thus - a measure of Shapley's value's failure to fully encode the game.

Lemma 2 then makes precise the relationship between the absence of interactions between variables and inessentiality.

Section 5 compares the Shapley residual to the Shapley-Taylor interaction index.

**Ethical Concerns:**

No concerns

**Limitations And Societal Impact:**

No concerns

**Main Review:**

**Originality**

There is a lot of work on variants of Shapley values, with interaction effects being one focus of that.  This paper's approach, to consider the 'poorness of fit' of a Shapley value is, in my experience, novel.

**Quality**

In general, I found the paper to be of high quality.

There are some minor errors which should be fixed:
- lines 5-6 should probably not claim that Shapley values don't approximate the _game_: they're not meant to; they're meant to be fair values that could arise from _playing_ the game.  Maybe something like: "Shapley values are a poor measure of interaction effects in the machine learning model, and corresponding game"?
- the example in lines 35-45 seems to get something wrong: we only have $f(g,d) > 0$ when $g$ and $d$ have _opposite_ signs.
- line 113: should be $v(S \cup i) - v(S)$, not $v(S \cup i) - u(S)$
- line 141: I think that this should read: "_If_ $v$ is not inessential..."

**Clarity**

The paper was well and clearly written.

I had trouble understanding the figures illustrating the paper's geometric methods.  I would therefore welcome suggestions from the authors about how these could be clarified.  Would illustrating the $n=2$ case be clearer?

**Significance**

Measures of feature importance are clearly of importance to the ML community, and the Shapley family leads among these.  Thus, trying to understand its limitations is an important task.  This paper provides novel insights into that.

The paper emphasizes its differences with the Shapley-Taylor interaction index.  The paper would have felt a bit more rounded if it also compared the Shapley residuals to the Grabisch-Roubens index, which kick-started the interaction index literature.

As the paper's project is ultimately a negative one - to reveal when Shapley value fails to capture interactions - it leaves open the question of what 'the right' interaction index might be.  As the Shapley residuals live in the same space as interaction indices, it's not clear that computing one of these should be more expensive.  I accept the reply, though, that a paper along these lines is a separate paper.

As a sequel, it could be informative for the authors to compare various measures of joint feature importance in light of their residuals.

**Time Spent Reviewing:**

4

---

> ### Author Response · Authors · 2021-08-10
> **Response to 6Qs2: Grabisch-Roubens index**
>
> Thank you for your helpful feedback and corrections. You pointed out that we should have discussed the Grabisch-Roubens (GR) index in the section where we describe Shapley-Taylor (ST) indices. We will include the citation, but we would argue that an in-depth treatment of its relationship to our work is unnecessary. The point of that discussion section is to argue that interaction indices have a fundamentally different objective from residuals, and there is little to be said about how they relate to Shapley residuals. The extent to which we can compare them is this: Like the kth order term of the Shapley-Taylor index, the GR index is based on an average of discrete derivatives. Discrete derivatives are affected by the interactions between players *within* a coalition; our residuals only capture interactions between a coalition and its *complement*. There is not a natural way to describe the nature of the relationship between residuals and interaction indices beyond this observation (and the fact that they both associate values to sets). We will add this brief explanation into the section in question.

---

### Official Review · Reviewer_YXMR · 2021-07-16

**Rating:** 7
**Confidence:** 4

**Summary:**

This paper introduces Shapley Residuals, a new measure that can be calculated in addition to SHAP values. This measure quantifies in some sense the information lost by Shapley Value due to interactions or correlations between features. This measure can help practitioners to identify scenarios when a score given to a certain feature by SHAP may not correctly quantify the effect of this feature on the prediction due to interactions or correlations. The paper provides theoretical grounding for the residuals, along with toy examples and an experiment with UCI dataset.


**Limitations And Societal Impact:**

I think that the authors addressed these topics well

**Main Review:**

Strength:

the paper is well written and provides a novel way to quantify the information lost by SHAP values, a well-known problem that has been investigated extensively in recent literature (https://arxiv.org/abs/2002.11097, https://arxiv.org/abs/1908.08474).

The authors provide a solid theoretical grounding for the new measure suggested and address the most common variants of SHAP. The paper contains several examples that are easy to understand and help motivate this work well. In addition, the authors describe in what sense the Shapley Residuals are different from existing feature interaction measures and also discuss the limitations of their work.

Weaknesses:
My main concern about this paper is its maturity which can mainly affect its impact. As the authors describe in the paper, the use and meaning of their new suggested measure are not 100% clear yet. In addition, the authors do not suggest an efficient approximation for calculating the residuals, which makes this method currently not feasible for datasets with more than ~20 features.

Moreover, I think that additional empirical results with real datasets might help to get a better understanding of how often these residuals can come into play (e.g as in https://arxiv.org/abs/2006.16234).

Overall, IMHO the novelty and theoretical grounding of this work are worth publish and may enable further research to overcome the challenges I discussed above.

Additional notes:
On line 39 it said that f(g, d) > 0 iff g and d are for the same sign. If I follow correctly, the opposite is true.
On the end of line 113 u(S) should be \nu(S)
Also in line 116 \nu should be used instead of u







**Time Spent Reviewing:**

5 hours

---

> ### Author Response · Authors · 2021-08-10
> **Response to YXMR: Experiments and algorithms**
>
> Thank you for your thoughtful comments and suggestions; we have taken the corrections into account. As we describe in our response to Reviewer Tdmx, we feel that the development of a fast approximation algorithm and the resulting large experiments it would enable are more appropriate for a future paper. This feeds into the second point about empirical results with real data sets. We’ve tried to illustrate some of the issues that arise with smaller (but real) data sets (see the appendix), but for extended experiments with larger sets we will need to build fast approximations.

---

> > ### Comment · Reviewer_YXMR · 2021-08-18
> > **YXMR Reviewer Final Response**
> >
> > Thanks for your response. I'm still a bit torn about leaving the approximation for future work, but not changing my score since eventually I think that this work is enough innovative to stand by its own as a paper.

---

### Official Review · Reviewer_tknB · 2021-07-16

**Rating:** 5
**Confidence:** 4

**Summary:**

The authors focus on the limits of the Shapley value-based explanation that cannot explain the synergy effect among more than two features. To resolve this, they provide a new measurement called Shapley residual, which measure excess of adding a feature to some set of features. Experimental results show that the Shapley residual can capture more information that the Shapley cannot explain.

**Limitations And Societal Impact:**

As described in the significance, a limitation of the paper is the definition of the Shapley residual, which may not evaluate the interaction of more than two players, which does not show that the Shapley residual works well on any cooperative games. Thus, I think the paper needs further generalization of the definitions.
In addition, as described in Conclusion section, the proposed spend exponential time in the number of players to find Shapley residual.


**Main Review:**

Originality:
It seems to be a new insight for the Shapley-based explanation to measure a difference from inessentiality. This idea is completely converse from the previous work by [Stern and Tettenhorst 2019], who tries to approximate a game as a combination of inessential games.

Quality:
 The proposed propositions seems to be technically sound and experimental results support claims of the paper.
 To my knowledge, the definition of the Shapley residual is new and the theoretical and experimental results are correct, but some theoretical results seem to be trivial, e.g., Proposition 1, and the definition does not seem to be sufficient to measure interactions among multi players (details are written in the significance).

Clarity:
Almost parts of the paper are well written and well-organized. I can easily follow the content.

Significance:
The definitions about Shapley residual (Definitions 3 and 4), that is, the definitions of $\Delta_S$ and $r_S$,  does not seem to be sufficient for generalizing the marginal value contributed by more than two players. When we compute $r_S$, it lacks to measure the interaction among players S because $r_S$ is the summation of $r_i$. An easy example that the definition does not work is a supermodular game (or convex game). When we consider to add i and j to C on a supermodular function v, evidently the marginal values of summation of i and j and ij holds $v(C \cup i) - v(C) + v(C \cup j) - v(C) \le v(C \cup ij) - v(C)$, which implies proposed definition underestimate the marginal contribution and seems not care about the interaction between i and j for adding to C. So, I think it would be better to modify the definition for evaluating more accurately.

Typo:
In Theorem 1, it would be better to use the same notation as this paper, that is, N should be written to [d].

**Time Spent Reviewing:**

6 hours

---

> ### Author Response · Authors · 2021-08-10
> **Response to tknB clarifying r_S**
>
> Thank you for your thoughtful comments and for catching that typo.
>
> We would like to address your concern about $r_S$. From our understanding of your argument, the property you describe with an example is not in fact a problem for our definition: in section 5, we explain that $r_S$ is *not* meant to capture interactions between the variables in S. Instead, it captures interactions between S and its *complement.* Since this property is what most intuitively distinguishes our work from the commonly-cited Shapley-Taylor index, we described this discussion as a “comparison with related work”. In hindsight, we could have provided this intuition earlier in the paper. We will update our next version to do so.
>
> With that said, if the analytical goal is to examine the existence of interactions between features i and j (as seems to be your goal given that comment), we want to make it clear that one *can* in fact use Shapley Residuals. Specifically, if the norms of $r_{i}$ and $r_{j}$ are large compared to the norm of $r_{\{i,j\}}$, this indicates that the majority of the interaction between i and other players happens with j (given that the interactions between the set $\{i,j\}$ and its complement are relatively small).
>
> Finally, we point out that the norms of $r_S$, etc. are indicators of the relative magnitude of interactions of S with its complement, rather than indicators of whether the interactions within S are destructive or constructive (which appears to be what your review implies from the supermodularity comment).

---

### Official Review · Reviewer_Tdmx · 2021-07-16

**Rating:** 4
**Confidence:** 4

**Summary:**

This paper presents Shapley residual as a measure of player interactions for the interpretation games arising in machine learing applications.  The definition of Shapley residual is based on a classical notion of ineseentiality in cooperative games.  Intuitively, it captures how much deviation a game is from being inessential (or modular in combinatorics) .

**Limitations And Societal Impact:**

No

Could discuss potential negative societal impact resulted from biases when interpreating model outputs using Shapley values and Shapley residuals.


-------------------
Change of the rate:
I have read the feedback for me and also for other reviews. Some of the comments were not properly addressed. Also considering theoretical novelty, now I think the paper is bellow the acceptance bar.

**Main Review:**

Originality & quality:

-  I like the idea of introducing Shapley residuals as complementary information for black-box model interpretation, also the theoretical foundation from game inessentiality is solid, though the proofs do not need lots of creativity.  As far as I know, this is the paper adapted from one paper of ICML 2020 Workshop on Human Interpretability in Machine Learning (WHI 2020).

- The algorithm for caculating Shapley residuals (Algorithm 1) needs exponential computational complexity (at least $2^n$),  it is better to explore a bit approximating algorithms, such as sampling based ones to have a speedup.



Clarity & experiments:

-  I like the illustrating examples  ($f_1, f_2$ and so on) along the main text. However, it is better to control the amount of illstration a bit, so you would have enough space for more technical details in the main text.

- Better to give a bit more detailed explanations of the experimental results.

-  Though it is computationally intractable to precisely compute Shapley values, one can still compute them for games with player number up to, say 25.  Would be nice to see these results other than results with around 10 players.


**Time Spent Reviewing:**

6

---

> ### Author Response · Authors · 2021-08-10
> **Response to Tdmx: Computational limitations**
>
> Thank you for your thoughtful comments. We agree with your assessment of our work’s computational limitations, and this is something we are actively working on. With that said, this full account of the computational machinery needed to overcome the performance limitations of the methods as presented here lies beyond the scope of the present submission; the theory required to justify the definition itself requires an exposition that takes all of the available space in the submission. A future paper with a more powerful algorithm would, indeed, enable larger experiments as you suggest; we have made promising steps towards this.

---

> ### Author Response · Authors · 2021-09-09
> **Which comments were not properly addressed?**
>
> I am not sure when you updated your review, because I just now noticed you changed your rating. Could you please specify which comments you thought were improperly addressed, and in particular why these comments caused you to now think this is below the acceptance threshold?

---

### Decision · Program_Chairs · 2021-09-27

**Decision:**

Accept (Poster)

**Comment:**

The paper presents a way  to detect when local feature importance scores might be providing misleading signal due to feature interaction. This is an interesting problem that was not addressed before. The techniques used are not surprising but are sound. The main limitations of this work are the missing discussion about computational complexity and the fact that an earlier version of this paper appeared in an ICML workshop.